# Consistent modelling of material weight loss and gas release due to pyrolysis and conducting benchmark tests of the model—A case for glovebox panel materials such as polymethyl methacrylate

**Takuya Ohno** *, **Shinsuke Tashiro, Yuki Amano, Naoki Yoshida, Ryoichiro Yoshida, Hitoshi Abe**

Japan Atomic Energy Agency, Tokai, Japan

* ohno.takuya@jaea.go.jp

**Data Availability Statement:** All relevant data are within the manuscript.

## Abstract

It is necessary to consider how a glove box's confinement function will be lost when evaluating the amount of radioactive material leaking from a nuclear facility during a fire. In this study, we build a model that consistently explains the weight loss of glove box materials because of heat input from a flame and accompanying generation of the pyrolysis gas. The weight loss suggests thinning of the glove box housing, and the generation of pyrolysis gas suggests the possibility of fire spreading. The target was polymethyl methacrylate (PMMA), used as the glove box panel. Thermal gravimetric tests on PMMA determined the parameters to be substituted in the Arrhenius equation for predicting the weight loss in pyrolysis. The pyrolysis process of PMMA was divided into 3 stages with activation energies of 62 kJ/mol, 250 kJ/mol, and 265 kJ/mol. Furthermore, quantifying the gas composition revealed that the composition of the pyrolysis gas released from PMMA can be approximated as 100% methyl methacrylate. This result suggests that the released amount of methyl methacrylate can be estimated by the Arrhenius equation. To investigate the validity of such estimation, a sealed vessel test was performed. In this test, we observed increase of the number of gas molecules during the pyrolysis as internal pressure change of the vessel. The number of gas molecules was similar to that estimated from the Arrhenius equation, and indicated the validity of our method. Moreover, we also performed the same tests on bisphenol-A-polycarbonate (PC) for comparison. In case of PC, the number of gas molecules obtained in the vessel test was higher than the estimated value.

## Introduction

When assessing the amount of radioactive material leaking from a nuclear facility during a fire, it is critical to consider how the containment function of the glove box will be lost. For example, if a fire is stopped early, radioactive material leakage will be minimized because the

**Funding:** This study was performed under the research entrusted by Secretariat of Nuclear Regulation Authority. The funders had no role in study design, data collection and analysis, decision to publish, or preparation of the manuscript.

**Competing interests:** The authors have declared that no competing interests exist.

containment barriers, such as the glove box, will still perform well in the early stages of the fire. Therefore, it is necessary to understand how containment barriers to radioactive material are lost as the fire progresses. The glove box is one such containment barrier in many nuclear facilities. Therefore, this study models the process by which the containment function of the glove box is lost during a fire.

Research on glovebox fires has been carried out vigorously from the 1960s to the 1970s. Overview of these studies have been summarized in a literature review by Hart [1]. The literature review suggests that research on glovebox fires has focused primarily on material selections and glovebox designs [2] or initial fire extinguishing in the event of a fire [3, 4]. In other words, research on how to prevent fires is the mainstream, and little research has been done on what kind of effects will occur in the event of a fire. Some studies have addressed the effects of fire, but the focus have been primarily on the thermal effects of combustion on the surroundings [5, 6]. On the other hand, recently, studies on how glove box fires progress were conducted by conducting actual combustion tests on full-scale glove boxes [7, 8]. In these studies, for example, one glove box was burned to investigate the release amount of heat and soot to examine how it affects the facility. However, the studies did not consider when the containment function of the glove box was lost. The Japan Atomic Energy Agency (JAEA) modeled the pyrolysis behavior of materials used in glove box panels by conducting thermogravimetric-differential thermal analysis (TG-DTA) tests and differential scanning calorimetry (DSC) tests [9–11] to estimate when the containment function of a glove box was lost. For example, the weight loss of the panel materials in TG-DTA tests is a reduction in the thickness of the glove box barrier during a fire. As an extension of JAEA's studies, this study was conducted.

In this study, we consider a scenario of a glovebox fire (Fig 1). We assume that the fire occurs outside the glove box in a process room, and the inside of the glove box is filled with an inert gas such as $N_2$, and the process room is filled with air. The fire heats the glove box. When the temperature of the glove box rises because of heating, the constituent materials of the glove box, especially polymers, such as organic glass and rubber, can be pyrolyzed, reducing the thickness of the glove box barrier. An indicator of this barrier thickness is the weight of the polymer. As the pyrolysis progresses, the weight of the polymer decreases, which is equivalent to making holes in the glove box during a fire. The mass lost by pyrolysis is released to the outside and inside of the glove box as a flammable pyrolysis gas. On the outside, the pyrolysis gas

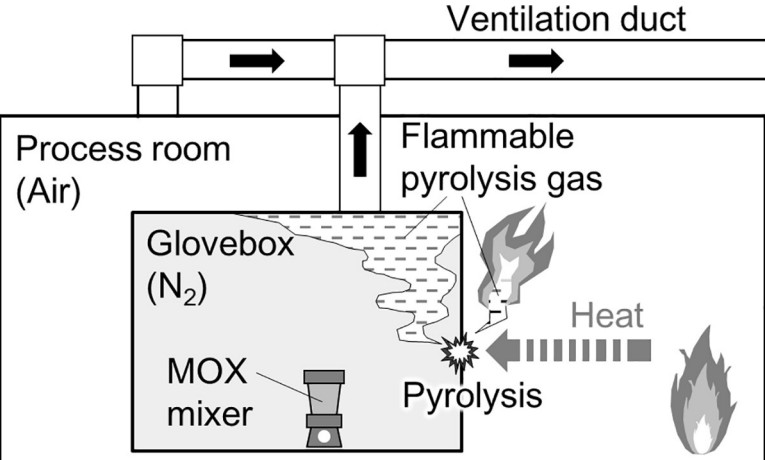

**Fig 1. Schematic of glovebox fire.** This study models the reduction of glovebox barrier thickness and release of flammable gas because of pyrolysis.

will mix with air and might burn. However, inside the glove box, the pyrolysis gas does not burn immediately because of the inert state. Therefore, pyrolysis gas accumulates in the glove box or migrates with the inert gas through the ventilation duct. If the glove box seal, secured by rubber-based materials such as chloroprene rubbers, is weakened and air in the process room flows into the glove box, the accumulated pyrolysis gas could explode and burn. Furthermore, the pyrolysis gas migrating through the ventilation ducts could burn when they merge with air, accelerating the pyrolysis of the polymer in the glove box. As the pyrolysis decreases the weight of the glove box, radioactive materials trapped inside will eventually leak into the process room, which is the moment when the containment function of the glove box is lost.

Modeling the process of losing the containment function of gloveboxes in fire cases requires investigating the behavior of the pyrolysis and flammable gas release of the glovebox components. Specifically, to closely follow the migration of released pyrolysis gas, each gas component contained in the gas must be investigated because each gas component has a different vapor pressure curve. For example, at a certain ambient temperature, some high-vapor-pressure components move as gas, whereas low-vapor-pressure components condense or solidify. Therefore, the overall composition of the pyrolysis gas changes according to the ambient temperature distribution. The composition of combustible gas is critical information because it affects the ignitability of the gas and the amount of heat released during combustion.

Therefore, this study is set to enable consistent evaluation of the weight loss of materials because of pyrolysis and the amount of each gas component generated. Furthermore, the behavior of the generated gas depends on the vapor pressure curve of each contained component; therefore, it is ideal to understand the gas generation amount for each component.

The material used in this study was polymethyl methacrylate (PMMA). PMMA is a flammable material used for glovebox panels. This material was set as the principal target in this study because it occupies most of the mass of combustibles present in the glove box. For comparison, we also investigated bisphenol-A-polycarbonate (PC), which is also used as a panel material. However, as mentioned above, it is thought that rubber-based materials are the most worrisome about glovebox seals. Note that this study is also a preliminary step aimed at establishing a methodology to begin the investigation of the rubber-based materials.

In this study, first, PMMA pyrolysis tests were performed by thermogravimetric -differential thermal analysis-mass spectrometry (TG-DTA-MS). Based on the test results, we modeled the progress of weight loss with changes in PMMA temperature. In this modeling, we conducted a kinetic analysis of TG data such as done in various fields [12–14]. This weight loss can be regarded as the total amount of pyrolysis gas released. Next, the breakdown of the total released amount for each gas component was investigated by conducting pyrolysis gas quantitative tests. The tests include analysis by pyrolysis-gas chromatography-mass spectrometry (Py-GC-MS) and experiments using a tubular furnace. The Py-GC-MS is an effective method for quantifying plastic materials and their decomposition products, and has been used in many studies [15–17]. Tubular furnaces have been also used as pyrolizers to support the quantifications [18]. The tests up to this point have made it possible to consistently evaluate the weight loss and amount of gas released from changes in the temperature of the PMMA or PC. Therefore, the purpose of this research was largely achieved. However, we thought it necessary to confirm whether this evaluation was appropriate. Therefore, we finally conducted sealed vessel tests. In the tests, a gram-order PMMA or PC in the sealed vessel was heated and pyrolyzed, and the change in temperature and internal pressure in the vessel because of gas generation were measured. We confirmed the validity of the model for estimating the release behavior of the pyrolysis gas obtained from the TG-DTA-MS tests by comparing the results calculated from the model with those measured from the sealed vessel tests.

## Materials and methods

### Materials

PMMA and PC were used as samples for the experiments described in the following sections. PMMA was manufactured by Mitsubishi Chemical Corporation (model number: ACRYLITE L001). The PC was manufactured by the C. I. TAKIRON Corporation (model number: 2000). The PMMA and PC were cut into small pieces of several millimeters and used as experimental samples to suit the TG-DTA-MS tests.

### TG-DTA-MS tests

TG-DTA-MS tests were performed to model the weight loss behavior of the sample because of pyrolysis. The analyzer used was manufactured by the Bruker Corporation (model number: TG-DTA2020SA / MS9610). This analysis used ~2.5 mg sample per run. The sample was placed on a platinum dish and heated under a nitrogen atmosphere at atmospheric pressure. The heating rate during heating was controlled as an experimental parameter at 5˚C, 10˚C, 15˚C, or 20˚C/min to kinetically analyze the behavior of sample weight loss because of pyrolysis, i.e., TG data, as described below. MS was performed by electron ionization with a scan range of $m/z$ = 2–200.

### Pyrolysis gas quantitative tests

Py-GC-MS analysis and tube furnace tests were performed to determine the composition of pyrolysis gas released from the sample. He was the carrier in all analyses and tests. The sample was heated at 10˚C/min. The released pyrolysis gas was collected in controlled temperature ranges of 150˚C-440˚C for PMMA and 440˚C-650˚C for PC.

The Py-GC-MS analysis was performed in two setups by combining one pyrolyzer and two GC-MS instruments. One setup consisted of a pyrolyzer manufactured by Frontier Laboratories Limited (model number: EGA / PY-3030D) and a GC-MS instrument manufactured by Shimadzu Corporation (model number: GCMS QP2010 Ultra). Another setup consisted of a pyrolyzer of the same model number and a GC-MS instrument manufactured by Agilent Technologies Incorporated (model number: GC / MS 5977B). Each Py-GC-MS analysis used a ~0.1 mg sample. The GC column was a weakly polar wall-coated open-tubular column manufactured by Frontier Laboratories (model number: UA5 (MS / HT) -15M-0.25F). The column was heated at 40˚C for 5 min after gas introduction and then to 320˚C at a rate of 15˚C/min. MS was performed in scan modes ranging from $m/z$ = 2–800.

The tube furnace tests were performed to quantify CO, $CO_2$, and $CH_4$ that were unsuitable for the Py-GC-MS analysis. In the test, ~0.3 g sample was heated in a tube furnace, and pyrolysis gas was collected in a gas bag by a He carrier gas. The collected gas was analyzed using two GC devices. The first was a Hitachi device (model number: G-3000) to quantify CO and $CH_4$, and the other, a different model from the same company (model number: 263–30), quantified $CO_2$.

### Sealed vessel tests

A sealed vessel test was performed as a gram-order scale benchmark test on the data obtained from the TG-DTA-MS and pyrolysis gas composition analyses. Fig 2 shows the setup of the sealed vessel test. In this test, a ~2.0 g sample was sealed in a 0.001 m³ cylindrical vessel filled with $N_2$ gas and heated at a rate of 10˚C/min. During heating, the temperatures of the sample and gas phase in the vessel were measured using thermocouples. The pressure change in the vessel was measured using a pressure gauge. The measured internal pressure was converted to

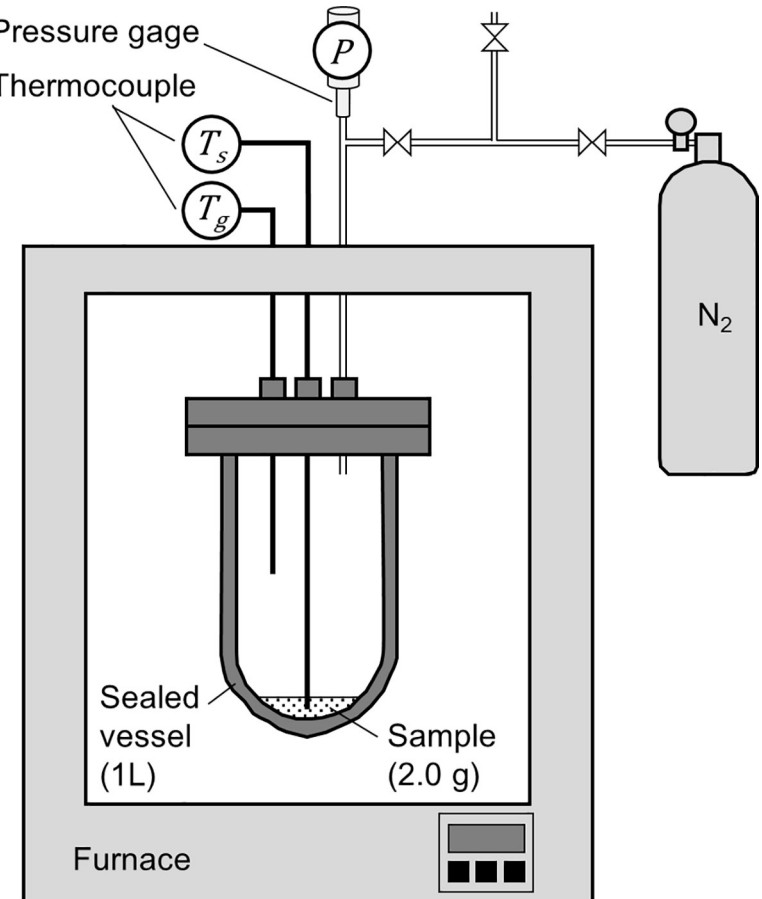

**Fig 2. Schematic of sealed vessel tests.** The increase in internal pressure because of pyrolysis gas release was measured.

the partial pressure of pyrolysis gas by subtracting the pressure measured in the blank ($N_2$ gas only) test. From the gas phase temperature and partial pressure of pyrolysis gas, the number of pyrolysis gas molecules in the vessel was calculated based on the ideal gas equation of state.

$$PV = nRT_g, \tag{1}$$

where $P$ is the internal pressure measured in the test (Pa), $V$ is the volume of the vessel ($m^3$), $n$ is the number of the pyrolysis gas molecules (mol), $R$ is the gas constant (Pa $m^3$ $K^{-1}$· $mol^{-1}$), and $T_g$ is the measured gas phase temperature (K).

## Results and discussion

### TG-DTA-MS tests

The solid line curve in Fig 3 shows the change in TG of PMMA obtained from the TG-DTA-MS analysis. This figure shows the result when the heating rate is 10˚C/min. The pyrolysis process began at 150˚C and ended at 440˚C. These pyrolysis behaviors are similar to those reported in the literature [19–21]. Based on the change in the TG curve, the pyrolysis reaction of PMMA was divided into three stages. Note that this stage classification is not based on the pyrolysis mechanism of the material but a classification for modeling the TG curve. The reduction amount of TG was 2 wt% in Stage 1, 15 wt% in Stage 2, and 80 wt% in Stage 3, and

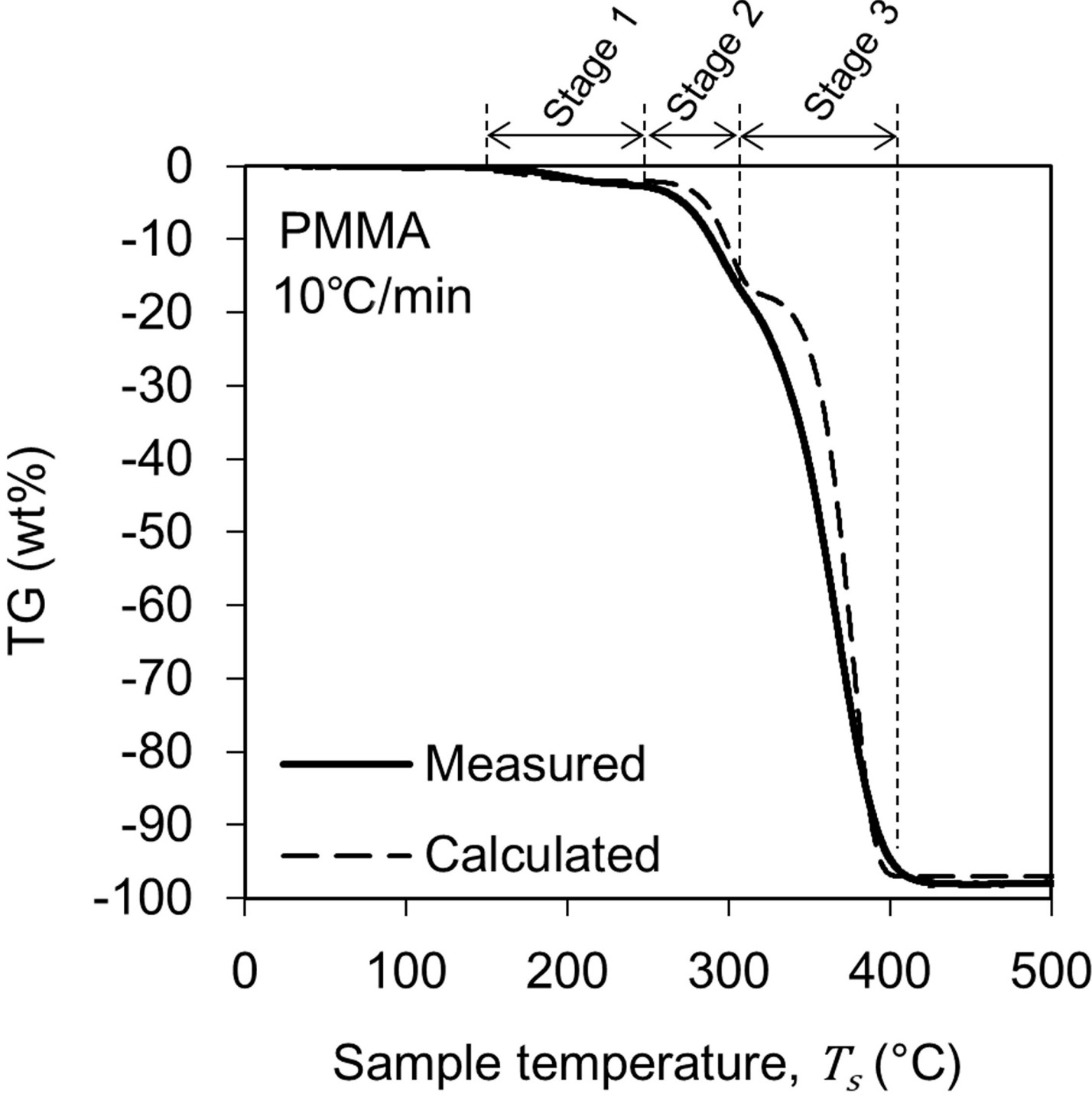

**Fig 3. Changes in TG during PMMA pyrolysis.** The solid curve shows the results of the TG-DTA-MS analysis. The broken curve is the result of the calculation based on Eq (4) and the parameters in Table 1.

the residue of pyrolysis was 3 wt%. These weight percentages represent the ratio of the initial weight of the sample. The TG reduction in Stage 1 may be correlated to release of methyl methacrylate which is contained in the PMMA as residual monomer and/or substances

absorbed on the surface of the sample. Moreover, the inflection point between Stages 2 and 3 have been often reported in the literature and have been associated with the formation of microstructures called as domains [22–24].

Fig 4 shows the change in TG of the PC with a solid line when the heating rate is 10˚C/min. Unlike PMMA, at least in this study, we treated the pyrolysis of PC as a one-step reaction. The one-step pyrolysis reduced TG by 74 wt%, and the residue was 26 wt%. The pyrolysis process started at 440˚C and ended at 650˚C. These pyrolysis behaviors were not significantly different from those reported in the literature [25–27].

The TG change behaviors in Figs 3 and 4 were modeled by performing a kinetic analysis based on the Friedman-Ozawa [28, 29] method. With this method, a reaction kinetic analysis can be performed on the TG curve measured using the heating method. In this method, the rate of pyrolysis is described by Eq (2).

$$\frac{d\alpha}{dt} = A \ exp\left(-\frac{E_a}{RT_s}\right), \tag{2}$$

where $\alpha$ is the ratio indicating the reaction progress, $t$ is the time (s), $A$ is the frequency factor (1/s), $E_a$ is the activation energy (J/mol), $R$ is the gas constant (J/(K, mol)), and $T_s$ is the temperature of the solid sample (K). Among these parameters, $A$ and $E_a$ are specific to respective reactions. Therefore, once $A$ and $E_a$ are calculated, $d\alpha/dt$ can be estimated under any $T_s$.

In conducting the kinetic analysis, we assumed three factors. i) The progress of pyrolysis of the sample can be divided into several reaction stages. ii) Each reaction stage consists of one elementary reaction. iii) All elementary reactions are accompanied by sample weight loss. Therefore, $d\alpha$ can be evaluated based on the change in sample weight, which is the change in TG. Therefore, in this study, $\alpha$ is defined as

$$\alpha = \frac{\Delta m_{TG,i}}{m^*_{TG,i}}, \tag{3}$$

where $\Delta m_{TG,i}$ is the weight loss of the sample from the start of reaction Stage $i$ to a specific point in the stage (g), and $m^*_{TG,i}$ is the total weight loss expected in the reaction Stage $i$ (g). This equation is combined with Eq (1).

$$\frac{d}{dt}\frac{\Delta m_{TG,i}}{m^*_{TG,i}} = A \ exp\left(-\frac{E_a}{RT_s}\right) \tag{4}$$

To determine $A$ and $E_a$ in these equations, an Arrhenius plot was created based on the logarithm of Eq (1).

$$\log_e \frac{d\alpha}{dt} = -\frac{E_a}{RT_s} + \log_e A \tag{5}$$

Fig 5 shows an example of an Arrhenius plot. This figure is an example of plotting the value of $d\alpha/dt$ at $\alpha = 0.75$. From the slope and intercept of the line connecting these plots, a set of $A$ and $E_a$ values were obtained. Other values of $A$ and $E_a$ were similarly obtained at $\alpha = 0.50$ and 0.25. These three value sets were averaged to calculate a set of $A$ and $E_a$ values (Table 1). Furthermore, based on the calculated $A$ and $E_a$ values, we reproduced the TG curve measured in the TG-DTA-MS analysis by calculation. The broken lines in Figs 3 and 4 show the results of the reproduction calculation, which correlates well with the actual measurement result. The calculated $E_a$ of the Stage 2 and 3 in PMMA pyrolysis were 250 kJ/mol and 265 kJ/mol. These $E_a$ values are slightly higher than the generally reported 150–200 kJ/mol [30–32], however, there is also a report that provide close $E_a$ value [33].

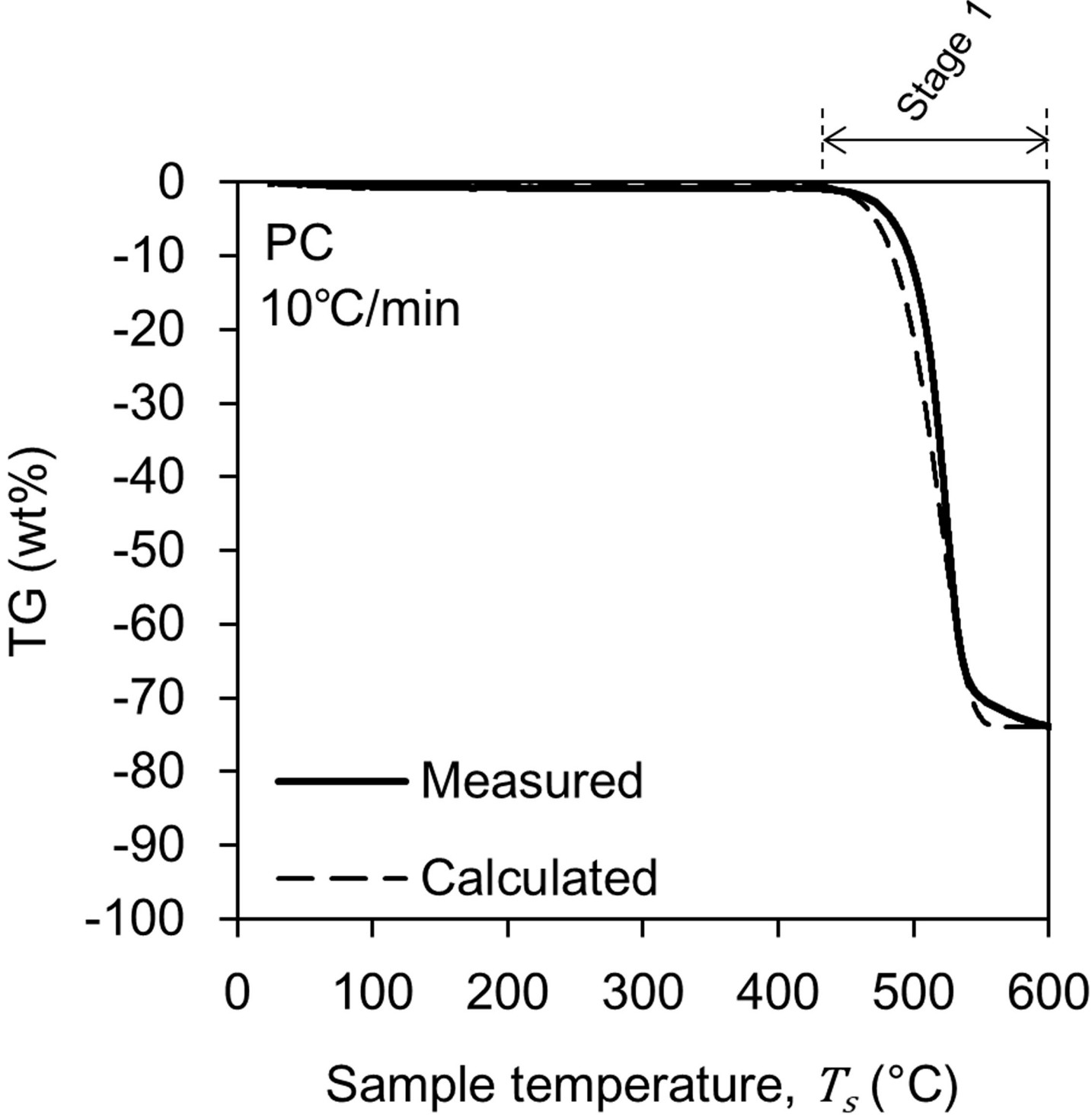

**Fig 4. Changes in TG during PC pyrolysis.** The solid curve shows the results of the TG-DTA-MS analysis. The broken curve is the result of the calculation based on Eq (4) and the parameters in Table 1.

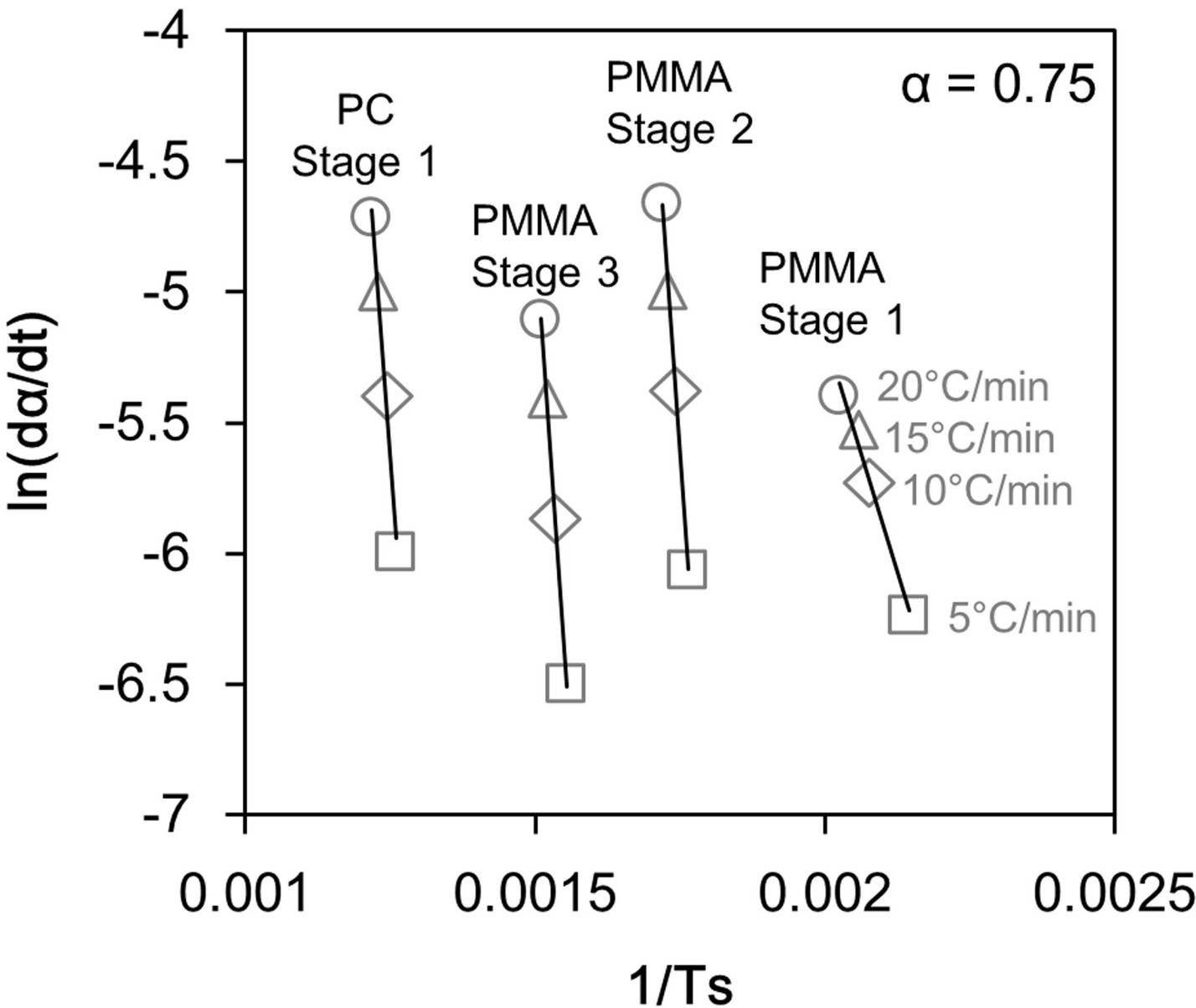

**Fig 5. An example of an Arrhenius plot for the results of TG-DTA-MS analysis.** From the slopes and intercepts of the approximate straight lines, $A$ and $E_a$ in Table 1 were determined.

**Table 1. Results of kinetic analysis for TG data.**

|  |  | PMMA | PMMA | PMMA | PC |
|---|---|---|---|---|---|
|  |  | Stage 1 | Stage 2 | Stage 3 |  |
| $A$ | (1/s) | $1.2 \times 10^5$ | $8.8 \times 10^{20}$ | $2.7 \times 10^{19}$ | $2.8 \times 10^{14}$ |
| $E_a$ | (kJ/mol) | 62 | 250 | 265 | 251 |
| $m_{TG,i}^*$ | (wt%)[a] | 2 | 15 | 80 | 74 |

[a] Relative to initial weight of the sample.

**Table 2. Results of pyrolysis gas composition analysis for PMMA.**

|  | Yield (wt%)[a] | $c_j$ (wt%)[b] | Measurement method |
|---|---|---|---|
| **Methyl methacrylate** | 97.7 | 100 | B |
| (TG residue) | 3 |  | E |
| **Total** | 100.7 | 100 |  |

B: Py-GC-MS (EGA/PY-3030D & GC/MS 5977B).

E: TG-DTA-MS (TG-DTA2020SA/MS9610).

[a] Relative to initial weight of the sample.

[b] Relative to total yield of quantified gas components.

## Pyrolysis gas quantitative tests

Tables 2 and 3 show the results of pyrolysis gas composition analysis. In the tables, the quantification results for each gas component are shown as wt% (yields) relative to the initial weight of the sample. The tables also show the wt% of the residue recovered in the TG-DTA-MS analysis. Ideally, the sum of all gas components and residues should be 100 wt%.

Table 2 shows the results for PMMA. Here, almost only methyl methacrylate was detected from the pyrolysis gas, like the example reported in the literature [34–36]. Methyl methacrylate is equivalent to the monomer material of PMMA. The total yield was 100.7 wt%, indicating that almost the entire amount was recovered.

Table 3 shows the results for PC. Here, the components detected were bisphenol-A and other lower aromatic compounds. Bisphenol-A is a substance equivalent to the monomer

**Table 3. Results of pyrolysis gas composition analysis for PC.**

|  | Yield (wt%)[a] | $c_j$ (wt%)[b] | Measurement method |
|---|---|---|---|
| **Bisphenol-A** | 20.3 | 34.5 | A |
| ***p*-Cumylphenol** | 0.6 | 1.1 | A |
| ***p*-tert-Butylphenol** | 0.7 | 1.1 | A |
| ***p*-Isopropenylphenol** | 1.0 | 1.7 | A |
| ***p*-Ethylphenol** | 5.2 | 8.9 | B |
| ***p*-Methylphenol** | 12.1 | 20.5 | B |
| ***p*-Xylene** | 0.3 | 0.5 | A |
| **Phenol** | 3.6 | 6.1 | B |
| **Toluene** | 1.0 | 1.7 | B |
| **Benzene** | 0.4 | 0.8 | B |
| **Methane** | 1.5 | 2.5 | C |
| **CO** | 1.1 | 1.8 | C |
| **$CO_2$** | 11.1 | 18.8 | D |
| (TG residue) | 26 |  | E |
| **Total** | 84.9 | 100 |  |

A: Py-GC-MS (EGA/PY-3030D & GCMS QP2010 Ultra).

B: Py-GC-MS (EGA/PY-3030D & GC/MS 5977B).

C: Tube furnace & GC (G-3000).

D: Tube furnace & GC (263–30).

E: TG-DTA-MS (TG-DTA2020SA/MS9610).

[a] Relative to initial weight of the sample.

[b] Relative to total yield of quantified gas components.

material for PC. The bisphenol-A have been typically measured and detected with similar Py-GC-MS analysis [37–39]. Moreover, the other species of detectives were like those reported in the literature [40–42]. In this study, the yield of bisphenol-A was 20.3 wt% for the initial weight of the sample. The total yield was 84.9 wt% because of some unidentified components.

Based on these results, hypothetical pyrolysis gas composition ratios $c_j$ (wt%) for each gas component $j$ were defined. PMMA assumed that the pyrolysis gas consisted only of methyl methacrylate. Therefore, the value of $c_j$ for methyl methacrylate was set to 100 wt%. For PC, unidentified components were ignored, and each yield was multiplied by a constant value so that the total yield of the quantified components was 100 wt%. The yields after multiplication were directly used as the values of $c_j$ for each gas component. Tables 2 and 3 show the $c_j$ values. These values were used in the discussion in the following section.

## Sealed vessel tests

Figs 6 and 7 show the results of the sealed vessel test. In the figures, the white plots show $n$, which is the number of pyrolysis gas molecules in the vessel calculated from Eq (1) based on the pressure measurement results.

Fig 6 shows the result for PMMA. Here, the value of measured $n$ increased at ~300˚C and reached a plateau at ~400˚C. This plateau is the endpoint of the pyrolysis of PMMA. After the plateau, the value of $n$ increased again from ~450˚C. The increase in $n$ after the plateau could indicate the pyrolysis of methyl methacrylate in the gas phase, of which there are reported examples [43].

Fig 7 shows the result for PC. Here, the measured $n$ increased significantly at ~400˚C, which is consistent with the pyrolysis stage defined by the TG-DTA-MS analysis in Fig 4. However, the endpoint of PC pyrolysis could not be properly determined because the

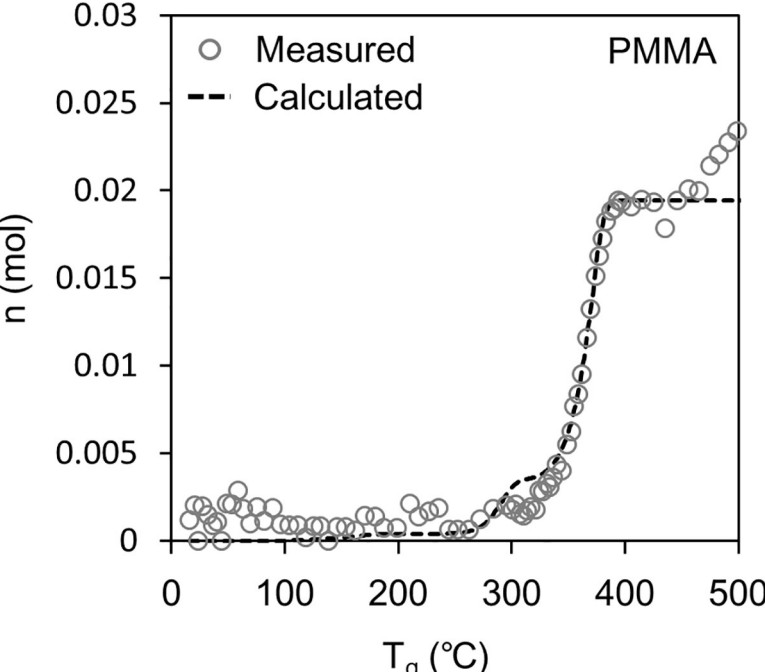

**Fig 6. Accumulation of pyrolysis gas in the container in the sealed container test for PMMA.** The white plot is calculated using Eq (1) from the measured inner pressure. The broken line is calculated using Eq (7) according to our model.

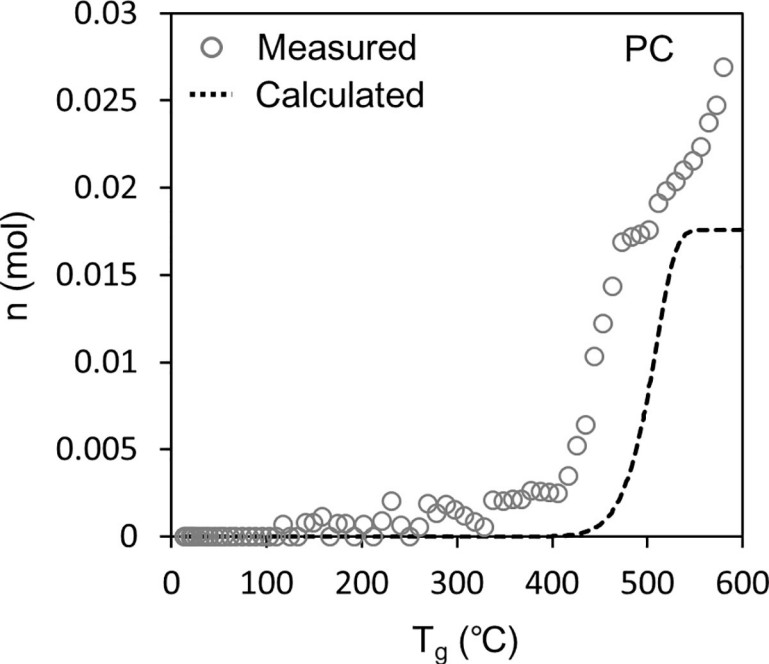

**Fig 7. Accumulation of pyrolysis gas in the container in the sealed container test for PC.** The white plot is calculated using Eq (1) from the measured inner pressure. The broken line is calculated using Eq (7) according to our model.

measured $n$ continued to increase by the end of the test, probably because the decomposition products released from PC continued to pyrolysis in the gas phase. For example, the pyrolysis temperature of bisphenol-A has been reported as 220°C [44] and 265°C [45].

We examined whether the results of the sealed vessel tests could be explained from the results of the TG-DTA-MS analysis and pyrolysis gas composition analysis. First, the weight loss $\Delta m_{TG,i}$ of the sample because of pyrolysis in the sealed container was calculated using Eq (4). For $A$ and $E_a$ in Eq (4), the values in Table 1 were substituted. For $T_s$ in Eq (4), the temperature of the sample measured in the sealed vessel test was substituted. The calculated $\Delta m_{TG,i}$ was regarded as the total amount of pyrolysis gas generated. From the total amount generated, the amount of each pyrolysis gas component was calculated using Eq (6).

$$m_{gas,j} = \frac{c_j}{100} \sum_i \Delta m_{TG,i}, \tag{6}$$

where $m_{gas,j}$ is the amount of each pyrolysis gas component (g). For $c_j$ in this equation, the values in Tables 2 and 3 were substituted. Then, the calculated $m_{gas,j}$ was converted into the amount of gas molecules, and the sum was calculated using

$$n = \sum_j \frac{m_{gas,j}}{M_j}, \tag{7}$$

where $M_j$ is the molecular weight of the gas component j (g/mol).

The dashed line in Figs 6 and 7 shows the $n$ calculated using Eq (7) and is compared to the $n$ measured using the method based on Eq (1). For PMMA, the calculated $n$ correlated well with the measured $n$, indicating the validity of the model constructed in this study. The model estimates the amount of each pyrolysis gas component released (Eq (4) and Eq (6)). However, for PC, the calculated $n$ shifted to a higher temperature than the measured $n$, which has not

been clarified so far; therefore, it is necessary for continuous studies in the future. One hypothesis is as follows: Pyrolysis in the gas phase could increase the number of gas molecules produced per weight loss of PC. Consequently, the pyrolysis speed of PC could be emphasized in Fig 7 compared to Fig 4.

## Conclusion

This research was conducted to consistently evaluate the weight loss of the glove box material and the number of gas components generated during pyrolysis. We conducted various tests on PMMA as a representative of glove box materials. For comparison, we also conducted a similar test on PC. The results obtained in this research are discussed below.

1. The TG-DTA-MS test determined the parameters of $A$ and $E_a$ in Eq (4). By using these parameters and Eq (4), we could calculate the weight loss of PMMA and PC in pyrolysis due to the temperature change of sample.

2. The pyrolysis gas quantitative tests revealed pyrolysis gas composition of PMMA and PC. Based on the results, we determined approximate composition ratio $c_j$ for each gas component. By multiplying $c_j$ with the weight loss of sample in pyrolysis, we could estimate release amount of each pyrolysis gas component.

3. The sealed vessel test verified that the method for estimating the gas release amount constructed above is appropriate. For PMMA, the method could well explain the gas generation in the sealed vessel test. This result shows that our method for estimating the gas generation amount is valid; therefore, the purpose of this research was achieved.

   In the future, we will continue research on the following:

1. It is necessary to examine conditions leading to combustion when each pyrolysis gas component contacts oxygen. It is also necessary to investigate whether the conditions for burning a mixed gas can be estimated by combining the conditions for burning each component.

2. The sealed vessel test of PC suggested that pyrolysis gas started to be generated earlier than expected in the TG-DTA-MS test. The cause has not been clarified, and the investigation should continue.

3. The sealed vessel test suggested that the pyrolysis gas generated from PMMA and PC could be further pyrolyzed in the gas phase in a high-temperature environment. Pyrolysis in the gas phase changes the ratio of combustible gas components. Therefore, the conditions causing pyrolysis in the gas phase must be investigated.

## Acknowledgments

The authors wish to thank Koji Watanabe and Shinichi Suzuki for technical support during the experiments and analysis; and all members of the Fuel Cycle Research Group of Japan Atomic Energy Agency for their contributions. Moreover, the manuscript of this paper was improved significantly based on comments of reviewers.

## Author Contributions

**Investigation:** Takuya Ohno.

**Methodology:** Takuya Ohno, Shinsuke Tashiro.

**Supervision:** Hitoshi Abe.

**Writing – original draft:** Takuya Ohno.

**Writing – review & editing:** Yuki Amano, Naoki Yoshida, Ryoichiro Yoshida.

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
