## [Decision Letter · Decision Letter 0]

25 Nov 2020

PONE-D-20-31208

Consistent modelling of material weight loss and gas release due to pyrolysis and conducting benchmark tests of the model　-　A case for glovebox panel materials such as polymethyl methacrylate

PLOS ONE

Dear Dr. Ohno

Thank you for submitting your manuscript to PLOS ONE. After careful consideration, we feel that it has merit but does not fully meet PLOS ONE’s publication criteria as it currently stands. Therefore, we invite you to submit a revised version of the manuscript that addresses the points raised during the review process.

We look forward to receiving your revised manuscript.

Kind regards,

Giuseppina Luciani

Academic Editor

PLOS ONE

Journal Requirements:

Reviewers' comments:

Reviewer's Responses to Questions

**Comments to the Author**

1. Is the manuscript technically sound, and do the data support the conclusions?

Reviewer #1: Partly

Reviewer #2: Yes

2. Has the statistical analysis been performed appropriately and rigorously? 

Reviewer #1: Yes

Reviewer #2: Yes

3. Have the authors made all data underlying the findings in their manuscript fully available?

Reviewer #1: Yes

Reviewer #2: Yes

4. Is the manuscript presented in an intelligible fashion and written in standard English?

Reviewer #1: Yes

Reviewer #2: Yes

5. Review Comments to the Author

Reviewer #1: This paper is interesting, and the experiment is well designed, but it lacks a more detailed analysis，Major revision is suggested.

1.Abstract: I think the author should summarize the key important information and results rather than expound the content of the experiment. The abstract should be concise and to the point. And line 43-44, the author did not point out the results of the comparison for the tests on bisphenol-A-polycarbonate.

2.In the introduction, it is necessary to add more literature to discuss the previous research of this work in order to further motivate the novelty of this research. Especially for thermogravimetry, the author can refer to the following literature: Journal of Hazardous Materials 402 (2021) 123472; Journal of Cleaner Production 239 (2019) 118113.

3.Line 25, What is the practical significance of this model when evaluating the amount of radioactive material leaking from a nuclear facility during a fire?

4.Line 43-44, “thermogravimetric (TG)-differential thermal analysis (DTA) (TG-DTA) tests and differential scanning calorimetry tests” should be corrected to “thermogravimetric-differential thermal analysis (TG-DTA) tests and differential scanning calorimetry (DSC) tests”. Similar mistakes were found in Line 115. Please check and revise them.

5.Line 137, The " ~2.5-mg" should be corrected to " ~2.5 mg", Similar mistakes were found through this paper. Please check and revise one by one.

6.Line 192, Could you explain why there are three stages in the pyrolysis of PMMA, and what substances are reacting in there three stages?

7.Line 189-197, 203-207, In this section, the pyrolysis behavior of PMMA / PC is lack of in-depth analysis.

8.Line 188, The analysis of kinetic is detailed, but there is a lack of relevant literature support, which can be referred to: Journal of Cleaner Production 253 (2020) 119950.

9.Conclusion: This part was too long. And the specific results discussion should appear in the "Results and Discussion" section and should not appear in the conclusion.

Reviewer #2: Revision of “Consistent modelling of material weight loss and gas release due to pyrolysis and conducting benchmark tests of the model - A case for glovebox panel materials such as polymethyl methacrylate”

Manuscript Number: PONE-D-20-31208

Review comments to the Authors: In this work glove box’s confinement function was studied in order to deeply investigate the issue related to the amount of radioactive material leaking from a nuclear facility during a fire. The authors built a model that consistently explained the weight loss of glove box materials (PMMA and PC based) because of heat input from a flame and accompanying generation of the pyrolysis gas, hence validating their calculation model through experimental data. I would like to congratulate authors for their exceptional research plan. They tried to study the aforementioned system by doing an array of examinations. The work is scientifically worthy of consideration, but technically can be further improved after a minor revision. I suggest authors consider the following points into account and revise their work:

• Lines: 78-80 - If the glove box seal is weakened and air in the process room flows into the glove box, the accumulated pyrolysis gas could explode and burn.

Reviewer: Could you specify which type of material as normally used as seal?

• Lines: 104-107 - The material used in this study was polymethyl methacrylate (PMMA). PMMA is a flammable material used for glovebox panels. This material was set as the principal target in this study because it occupies most of the mass of combustibles present in the glove box.

Reviewer: Is there any other reference in the literature reporting the study of the containment function of the glovebox based on PMMA polymer with a different configuration for the system?

• Lines: 112-116 - This weight loss can be regarded as the total amount of pyrolysis gas released. Next, the breakdown of the total released amount for each gas component was investigated by conducting pyrolysis gas quantitative tests. The tests include analysis by pyrolysis (Py)-gas chromatography (GC)-MS (Py-GC-MS) and experiments using a tubular furnace.

Reviewer: Similar measurements and experiments were recently published in some references (Primpke, Sebastian, et al. "Comparison of pyrolysis gas chromatography/mass spectrometry and hyperspectral FTIR imaging spectroscopy for the analysis of microplastics." Analytical and Bioanalytical Chemistry (2020): 1-16; Steinmetz, Zacharias, et al. "A simple method for the selective quantification of polyethylene, polypropylene, and polystyrene plastic debris in soil by pyrolysis-gas chromatography/mass spectrometry." Journal of Analytical and Applied Pyrolysis (2020): 104803; Yamane, Shogo, et al. "A data mining method from pyrolyzed products: Pyrolysis-gas chromatography-photoionization-high resolution time-of-flight mass spectrometry and kendrick mass defect analysis for polymer semiconductor poly (3-hexylthiophene)." Journal of Analytical and Applied Pyrolysis 151 (2020): 104923; Bifulco, Aurelio, et al. "Fire and mechanical properties of DGEBA-based epoxy resin cured with a cycloaliphatic hardener: Combined action of silica, melamine and DOPO-derivative." Materials & Design (2020): 108862; Bifulco, Aurelio, et al. "Improving flame retardancy of in-situ silica-epoxy nanocomposites cured with aliphatic hardener: Combined effect of DOPO-based flame-retardant and melamine." Composites Part C: Open Access 2 (2020): 100022). The authors should consider reading and citing them during their evaluations along the paper and this introduction section.

• Lines: 139-141 - The heating rate during heating was controlled as an experimental parameter at 5 °C, 10 °C, 15 °C, or 20 °C/min to kinetically analyze the behavior of sample weight loss because of pyrolysis, i.e., TG data, as described below.

Reviewer: Could you specify the ramp temperature at which the TGA measurement was performed?

• Lines: 193-196 - The reduction amount of TG was 2 wt% in Stage 1, 15 wt% in Stage 2, and 80 wt% in Stage 3, and the residue of pyrolysis was 3 wt%. These weight percentages represent the ratio of the initial weight of the sample. The pyrolysis process began at 150 °C and ended at 440 °C.

Reviewer: How do the authors explain these values for the weight losses along the pyrolysis process? Can you describe how the system appears at each stage from a chemical point of view?

• Lines: 245-247 - Fig 5. An example of an Arrhenius plot for the results of TG-DTA-MS analysis—from the slopes and intercepts of the approximate straight lines, and in Table 1 were determined.

Reviewer: Final dot for sentence 247 is missing.

• Lines: 281-283 - Bisphenol-A is a substance equivalent to the monomer material for PC. The species of detectives were like those reported in the literature [17–19].

Reviewer: Some references could be cited and inserted on sentence 283 for better supporting the importance of Py-GC/MS in the Bisphenol A measurement and detection (Becerra, Valentina, and Jürgen Odermatt. "Detection and quantification of traces of bisphenol A and bisphenol S in paper samples using analytical pyrolysis-GC/MS." Analyst 137.9 (2012): 2250-2259; Navarro-González, Rafael, Patrice Coll, and Roustam Aliev. "Pyrolysis of γ-irradiated bisphenol-A polycarbonate." Polymer Bulletin 48.1 (2002): 43-51; Bifulco, Aurelio, et al. "Fire and mechanical properties of DGEBA-based epoxy resin cured with a cycloaliphatic hardener: Combined action of silica, melamine and DOPO-derivative." Materials & Design (2020): 108862).

• Lines: 318-319 - For example, the pyrolysis of bisphenol-A occurs at 265 °C or higher [21].

Reviewer: I would suggest adding more references on sentence 319.

6. PLOS authors have the option to publish the peer review history of their article (what does this mean?). If published, this will include your full peer review and any attached files.

Reviewer #1: No

Reviewer #2: No

---

## [Author Response · Author response to Decision Letter 0]

21 Dec 2020

Dear Dr. Giuseppina Luciani

Thank you very much for your advice on our manuscript and for handling the peer review process during your busy schedule. We have revised the manuscript with reference to meaningful comments.

Following the email instructions, we uploaded the files named "Manuscript", "Revised Manuscript with Track Changes" and "Response to Reviewers" to the system. Please refer to these files for our response to technical comments from editors and reviewers. As part of the response to the comments, Fig. 3, 4 and 5 have been modified, so new files have been uploaded to the system.

Also, all figure files were uploaded to the PACE digital diagnostic tool and passed the check. At that time we made some modifications to Fig 1. Thus, we also uploaded a new file of Fig 1 to the system.

We would appreciate it if you could give our regards to the reviewers.

Best regards,

Takuya Ohno

Researcher, Nuclear Safety Research Center

Japan Atomic Energy Agency

2-4 Shirakata, Tokai-mura, Naka-gun, Ibaraki 319-1195, Japan

ohno.takuya@jaea.go.jp

---

## [Decision Letter · Decision Letter 1]

26 Dec 2020

Consistent modelling of material weight loss and gas release due to pyrolysis and conducting benchmark tests of the model　-　A case for glovebox panel materials such as polymethyl methacrylate

PONE-D-20-31208R1

Dear Dr. Ohno

We’re pleased to inform you that your manuscript has been judged scientifically suitable for publication and will be formally accepted for publication once it meets all outstanding technical requirements.

Kind regards,

Giuseppina Luciani

Academic Editor

PLOS ONE

Reviewers' comments:

Reviewer's Responses to Questions

**Comments to the Author**

1. If the authors have adequately addressed your comments raised in a previous round of review and you feel that this manuscript is now acceptable for publication, you may indicate that here to bypass the “Comments to the Author” section, enter your conflict of interest statement in the “Confidential to Editor” section, and submit your "Accept" recommendation.

Reviewer #1: All comments have been addressed

Reviewer #2: All comments have been addressed

2. Is the manuscript technically sound, and do the data support the conclusions?

Reviewer #1: Yes

Reviewer #2: Yes

3. Has the statistical analysis been performed appropriately and rigorously? 

Reviewer #1: Yes

Reviewer #2: Yes

4. Have the authors made all data underlying the findings in their manuscript fully available?

Reviewer #1: Yes

Reviewer #2: Yes

5. Is the manuscript presented in an intelligible fashion and written in standard English?

Reviewer #1: Yes

Reviewer #2: Yes

6. Review Comments to the Author

Reviewer #1: The authors have adequately addressed the comments raised in a previous round of review and this manuscript can be acceptable for publication.

Reviewer #2: The vast majority of comments received to date have been addressed, and the fixes have been incorporated into later releases of data. Taking into accoiunt the revised version of the manuscript, it can be accepted for publication in PLOS ONE.

7. PLOS authors have the option to publish the peer review history of their article (what does this mean?). If published, this will include your full peer review and any attached files.

Reviewer #1: No

Reviewer #2: No

---

## [Editor Report · Acceptance letter]

13 Jan 2021

PONE-D-20-31208R1 

Consistent modelling of material weight loss and gas release due to pyrolysis and conducting benchmark tests of the model　-　A case for glovebox panel materials such as polymethyl methacrylate 

Dear Dr. Ohno:

I'm pleased to inform you that your manuscript has been deemed suitable for publication in PLOS ONE. Congratulations! Your manuscript is now with our production department. 

Kind regards, 

on behalf of

Dr. Giuseppina Luciani 

Academic Editor

PLOS ONE